# The association of POSTN with postoperative recurrence risk in early-stage lung adenocarcinoma: From gene networks to cellular functions

Youde Xiao[1☯], Xuelian Lin[1☯], Hui Gong[2], Juan Hu[3,4*], Dan Wang[5*]

**1** Department of Oncology, Taikang Tongji (Wuhan) Hospital, Wuhan, China, **2** Department of Radiology, Wuhan Hospital of Traditional Chinese Medicine, Wuhan, China, **3** Lianyungang Second People's Hospital Affiliated to Kangda College of Nanjing Medical University, Lianyungang, China, **4** Lianyungang Clinical College, Bengbu Medical University & The Second People's Hospital of Lianyungang, Lianyungang, China, **5** Department of Radiology, Taikang Tongji (Wuhan) Hospital, Wuhan, China

☯ These authors contributed equally to this work.
* hujuan@lygey.com (JH); 337914199@qq.com (DW)

## Abstract

### Purpose

Periostin (POSTN) has been identified as a biomarker highly correlated with the risk of recurrence of lung adenocarcinoma (LUAD). Increasing the understanding of POSTN's role in LUAD recurrence may facilitate the development of more effective clinical interventions.

### Methods

Firstly, Weighted Gene Co-expression Network Analysis (WGCNA) analysis was used to identify key genes related to LUAD disease recurrence. Subsequently, the diagnostic performance of these genes was evaluated using Receiver Operating Characteristic (ROC) curves and validated using external datasets. The most effective genes were then analyzed using single- and multivariate Cox regression, linear regression, and stratified analysis to identify their clinical effects. Threshold effect analysis was used to identify the optimal threshold for POSTN expression, and an in vitro experiment was conducted to evaluate the effect of POSTN on tumor cell growth and invasion.

### Results

WGCNA analysis identified four genes strongly associated with LUAD recurrence. POSTN demonstrated good prediction performance for disease recurrence in both The Cancer Genome Atlas (TCGA) and GSE31210 datasets (Area Under the Curve (AUC)=0.693 and 0.743, respectively). Furthermore, POSTN was identified as an independent risk factor for disease recurrence in univariate and multivariate Cox

**Data availability statement:** All relevant data are within the manuscript and its Supporting information files.

**Funding:** This study was supported by the TaiKang Healthcare Research Fund (Youth Incubation Program) (2024008) awarded to Y.X., Lianyungang Science and Technology Bureau 2023 Municipal Key Research and Development Project (Social Development) (SF2321) awarded to J.H., Lianyungang Anti-Cancer Association 2023 Tumor Prevention and Treatment Science and Technology Development Plan Surface Project (MS202308) awarded to J.H., Nanjing Medical University Kangda College 2023 Key Research and Development Fund Project (KD2023KYJJ045) awarded to J.H., and Lianyungang 2024 hygiene and health Surface Science and Technology Project (202422) awarded to J.H.

**Competing interests:** The authors have declared that no competing interests exist.

regression analysis, and its expression level was found to be linearly related to disease recurrence in different populations. Threshold effect analysis indicated that a POSTN expression level of 12.5 was associated with a > 50% risk of recurrence within 1 year. In vitro experiments demonstrated that POSTN silencing could inhibit tumor cell proliferation and invasion.

## Conclusion

POSTN is positively correlated with the risk of early LUAD recurrence within 1 year after treatment, indicating its potential as a biomarker for predicting recurrence.

---

## 1 Introduction

Tumor progression in lung adenocarcinoma (LUAD) is a gradual process characterized by tumor growth, distant metastasis, lymph node enlargement, and postoperative recurrence. Curative resection is the standard treatment for stage I-II non-small cell lung cancer (NSCLC) [1]. The 5-year overall survival (OS) rate for stage I NSCLC patients after surgical resection is 66%−82%, while that for stage II NSCLC patients is 47%−52% [2]. The main factor leading to a decrease in survival rate is recurrence [3,4]. Among NSCLC patients who have undergone complete resection, 23.9% experience local recurrence or distant metastasis within 1 year [5]. The probability of recurrence within 5 years after surgery ranges from 20% to 75% [6]. Although adjuvant chemotherapy reduces the risk of recurrence-related death in lung cancer [7,8], not all patients receiving curative resection derive benefit. Pathological evaluation of surgical specimens can assess the risk of disease progression, such as lymph node metastasis [9], pleural invasion [10], lymphatic vessel invasion [11], vascular invasion [12], lepidic pattern adenocarcinoma subtype [13], and spreading through air spaces (STAS) [14,15]. However, subjective judgments of conventional prognostic factors hinder accurate prediction of recurrence and retrospective validation.

The widespread use of gene testing techniques has provided important support for the treatment decision-making and management strategies of patients with lung adenocarcinoma. In an observational study involving 426 LUAD patients [16], a predictive model was developed by integrating genomic and clinical pathological factors. This model (AUC: 0.73) outperformed TNM-based models (AUC: 0.55–0.61) in predicting recurrence rates. Fan Kou et al. [17] investigated different gene expression levels in lung adenocarcinoma and normal tissues in various GSE datasets, and further identified key genes involved in LUAD generation and progression through the STRING database and the Hubba plugin. The study showed that topoisomerase IIA (TOP2A) was highly expressed in lung adenocarcinoma and its expression level was negatively associated with the prognosis of LUAD patients. Lecai Xiong et al.[18] based on RNA sequencing data obtained from The Cancer Genome Atlas (TCGA) and validation results from the Gene Expression Omnibus (GEO) database, identified that the angiotensin II receptor 1 (AGTR1) encoding gene AGTR1 was associated with a better prognosis in LUAD and was a potential tumor suppressor gene.

Despite the identification of potential biomarkers for recurrence in lung adenocarcinoma patients in numerous studies, the reliability of the results is limited due to the lack of precise population focus and stratified analysis. For example, the analysis combines stages 1–4 patients and the lack of analysis in different populations to determine the generalizability of the biomarker for recurrence identification. Therefore, we aim to screen early-stage LUAD patient tissue sequencing data to identify key biomarkers for disease progression, providing a reference for identifying high-risk individuals for disease progression in clinical practice.

## 2 Materials and methods

### 2.1 Data sources and preprocessing

Data for stage I & II lung adenocarcinoma patients from The Cancer Genome Atlas (TCGA) were obtained from the UCSC Xena website (https://xena.ucsc.edu/). Patients with a progression-free interval (PFI) >5 years were defined as progression-free, while those with PFI <1 year were defined as high-risk for disease progression. GSE31210 (containing information on stage I & II lung adenocarcinoma patients who underwent surgery) and GSE166722 (including tumor invasion assessment information) were obtained from the Gene Expression Omnibus (GEO) database (https://www.ncbi.nlm.nih.gov/geo/). Patients with a postoperative recurrence period >5 years were defined as progression-free, while those with a recurrence period <1 year were defined as high-risk for disease progression. The inclusion and analysis purposes of various data, please refer to S2 Table.

### 2.2 Gene co-expression network construction identification of candidate hub genes

For TCGA data, genes with expression variance in the top 75% were selected to construct the gene co-expression network. The blockwiseModules function was used to convert the topological overlap matrix, with the following parameters: minClusterSize = 30 and power = 4. The plotDendroAndColors function was then used to construct and visualize the dendrogram, and the dynamic cutting algorithm was applied to merge the branches of the dendrogram into different gene modules. Different gene modules were represented by different colors. Genes that could not be assigned to any module were classified into a gray module and were removed from subsequent analysis.

Pearson's correlation test was performed to assess the correlation between modules and features of disease progression. This helped identify target modules. The correlation between genes within each module and clinical features of disease progression, as well as the correlation between genes and the module, was calculated to identify candidate hub genes.

### 2.3 Validation of hub genes

Firstly, in TCGA data, the prediction efficiency of hub genes was compared using the ROC curve and Kaplan-Meier (KM) curve with disease progression as the outcome variable. Secondly, in GSE31210 data, the prediction efficiency of hub genes was compared using the ROC curve and KM curve with postoperative recurrence as the outcome variable. Finally, in GSE166722 data, the prediction efficiency of hub genes was compared using the ROC curve and observing the expression differences between groups with invasive and non-invasive clinical information as the outcome variable. It should be noted that for KM analysis across all datasets (including TCGA-LUAD, GSE31210, and GSE166722), samples were divided based on the median value of POSTN expression: samples with POSTN expression below the median were defined as the "POSTN low-expression group", and those with POSTN expression above or equal to the median were defined as the "POSTN high-expression group".

### 2.4 Clinical efficacy analysis of hub genes

The clinical data of patients from TCGA and GSE31210 were combined to perform univariate and multivariate Cox proportional hazards regression analysis to determine the independent risk factors of key genes. Additionally, stratified analysis

was performed according to gender, age, and pathological stage to observe the consistency of the relationship between gene expression and disease progression in different populations. The generalized additive model and smooth curve fitting analysis were used to investigate whether there was a linear relationship and threshold effect between key genes and disease progression in different populations.

## 2.5 Functional enrichment analysis

The key genes were input into Genemania (http://www.genemania.org) to obtain the co-expression gene network and potential functions of the key genes. The clusterProfiler software package in R software was used to perform Gene Ontology (GO) analysis and Gene Set Enrichment Analysis (GSEA) analysis.

## 2.6 Cell culture and treatment

The H1975 and A549 lung adenocarcinoma cell lines were purchased from the Chinese Academy of Sciences cell bank. All cell lines were cultured with complete DMEM (Dulbecco's Modified Eagle Medium) medium containing 10% heat-inactivated fetal bovine serum and 100 U/ml penicillin/streptomycin, and incubated at 37°C in a humidified incubator containing 5% $CO_2$. Jiema Genes designed and produced POSTN-targeting siRNA (siPOSTN) and negative control siRNA containing POSTN targeting. During transfection, A549 and H1975 cells were planted in 6-well plates and allowed to adhere overnight. After 6 hours of transfection, normal culture medium was used to replace the transfection medium. When the cell density reached 80%, cells could be harvested for subculture or further experimentation.

## 2.7 Plasmid transfection

A549 and H1975 cells in good growth condition were seeded at a density of $2 \times 10^5$ cells per well in 6-well cell culture plates. Once the cell confluence reached over 70%, plasmid transfection was performed following the Lipofectamine 2000 transfection reagent kit instructions. Specifically, Overexpression Negative Control (OE_NC) and Overexpression of POSTN (OE_POSTN) plasmids and Lipofectamine 2000 were each dissolved in serum-free DMEM medium. For each well, 2 μg of plasmid and 6 μL of Lipofectamine 2000 were mixed in 100 μL of serum-free DMEM. After combining the plasmid and Lipofectamine 2000 solutions, the mixture was incubated at room temperature for 15 minutes before being gently added to the cell culture medium. After 6 hours, the medium was replaced with fresh DMEM containing 10% Fetal Bovine Serum (FBS).

## 2.8 Wound healing experiment

The logarithmic growth phase human non-small cell lung cancer cells A549 and human lung adenocarcinoma cells H1975 were digested and inoculated into a 6-well plate. After the cells were fully seeded, a cell scratch was made with a 200 μL pipette tip perpendicular to the well plate to ensure consistent scratch widths. The cell culture medium was aspirated and replaced with Phosphate-Buffered Saline (PBS) to remove cell fragments caused by the scratch. Serum-free culture medium was added, and photos were taken to record the experiment. The plate was then placed in the incubator, and photos were taken every 24 hours. Finally, the experimental results were analyzed based on the collected images.

## 2.9 Clone formation experiment

The logarithmic growth phase A549 and H1975 cells were suspended and inoculated into a 6-well plate (3000 cells per well). The cells were cultured with DMEM medium and changed every 3 days. Clone formation was stopped after 2 days of continuous culture. The culture medium was removed, and the cells were fixed with methanol for 20 minutes at room temperature. The cells were then stained with 0.1% crystal violet for 20 minutes to fix the clones. The results were photographed and counted.

## 2.10 Transswell invasion experiment

The invasive ability of A549 and H1975 cells was assessed using the Transwell assay. A 1% BSA solution was prepared using serum-free DMEM culture medium, and A549 and H1975 cells were cultured in the Transwell upper chamber with the 1% BSA solution. A layer of basement membrane matrix was applied to the upper side of the polycarbonate membrane to mimic the extracellular matrix. The lower chamber was filled with DMEM culture medium containing 10% fetal bovine serum. After 24 hours of incubation at 37°C, the cells in the upper chamber were removed, and the cells in the lower chamber were photographed. Five random staining results were photographed under a 200x magnifying glass.

## 2.11 Statistical analysis

All analyses were performed R software version 4.2.1 (http://www.Rproject.org) and EmpowerStats software (http://www.empowerstats.com). The Wilcoxon.test was used to calculate the significance of differences between two groups of patients. The "ggplot2" package in R was used for Kaplan-Meier survival analysis and log rank test to evaluate the survival status of different groups. The "pROC" package in R was used for receiver operating characteristic (ROC) curve analysis to assess the accuracy and sensitivity of our genes in diagnosing disease progression. Cytological experiments were statistically analyzed using GraphPad Prism v90 (GraphPad Software, La Jolla, CA, USA, http://www.graphpad.com). The two-tailed Student's t-test was used to evaluate statistical significance, and the asterisk in the figure represents the significance level, $P < 0.05$ was considered statistically significant.

# 3 Results

## 3.1 Data download and acquisition

Data information of early-stage (stage I & II) lung adenocarcinoma patients from The Cancer Genome Atlas (TCGA) was successfully obtained from the UCSC Xena website, including transcriptomic data and clinical information for 84 patients (comprising 56 cases of disease progression and 28 cases without disease progression). Additionally, datasets of early-stage LUAD patients (stage I & II) were downloaded from the GEO database, including GSE31210 (n = 226, with 64 cases of recurrence and 162 cases without recurrence) and GSE166722 (n = 53, with 21 cases of high invasiveness and 32 cases of dormancy), which provided mRNA expression profiles and clinical information. The research workflow is illustrated in Fig 1.

## 3.2 Identification of key genes associated with disease progression

Based on the TCGA data of the selected 84 patients, genes were ranked according to their expression differences, and the top 75% of genes were further analyzed. A soft-thresholding power of $\beta = 4$ was chosen for constructing the scale-free network (Fig 2A, B). These genes were then clustered into 16 modules based on average linkage hierarchical clustering (Fig 2C). By integrating clinical features, a positive correlation was observed between the clinical features of disease progression and the red module (R = 0.32, P = 0.004) (Fig 2D). The scatter plot demonstrated the significance of disease progression with the module membership of genes in the red module, indicating a significant positive association between the red module and disease recurrence (Fig 2E). Finally, four key genes (SNAI2, ARSI, GREM1, POSTN) associated with disease progression were identified based on the criteria of GS > 0.3 and MM > 0.8.

## 3.3 Screening and validation of key genes associated with cancer progression

The predictive performance of the four genes in disease progression was evaluated using ROC curves in the TCGA database, with AUC values of POSTN: 0.693, GREM1: 0.662, SNAI2: 0.714, and ARSI: 0.659 (Fig 3A). In terms of expression differences, all four genes showed significantly higher expression in the disease progression group (Fig 3B). Further evaluation was conducted on the correlation between the expression of these four genes and prognosis. The results showed that patients with high expression of POSTN and SNAI had poor prognosis (Fig 3C).

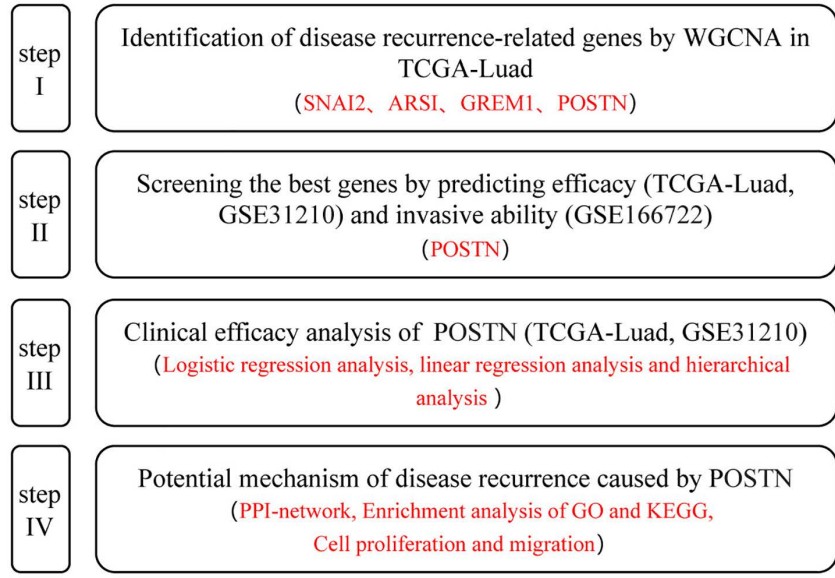

**Fig 1. Study flowchart.**

Subsequently, external validation was performed using the GSE31210 dataset. The AUC values for predicting postoperative recurrence for the four genes were as follows: POSTN: 0.743, GREM1: 0.730, SNAI2: 0.654, and ARSI: 0.543 (Fig 4A). Among the recurrent cases, except for ARSI, the other three genes showed significantly higher expression (P < 0.05) (Fig 4B). In addition, we investigated the association between the expression of these four genes and overall survival, and the results showed that, except for ARSI (P > 0.05), patients with high expression of the other three genes had poor prognosis (Fig 4C).

Finally, the GSE166722 dataset was used to evaluate the predictive performance of the four genes in invasion. The AUC values for the four genes were as follows: POSTN: 0.935, GREM1: 0.888, SNAI2: 0.757, and ARSI: 0.655 (Fig 4D). In the subsequent subgroup comparison, except for ARSI, the other three genes (P < 0.05) exhibited significantly higher expression in the high invasiveness group (Fig 4E).

### 3.4 Clinical efficacy analysis of key gene POSTN

Based on the previous analyses, POSTN was identified as a key biomarker for further analysis. The baseline demographic and clinical characteristics of patients in the TCGA dataset (n = 84, including 56 cases of disease progression and 28 cases without progression) and the GEO dataset (n = 105, including 17 cases of progression and 88 cases without progression) were analyzed (Table 1). Significant statistical differences were observed in POSTN expression and disease analysis between the two groups (P < 0.001), while no statistical differences were found in patient gender and age (P > 0.05).

Furthermore, a univariate Cox regression analysis was performed to assess the association of gender, age, disease stage, and POSTN with lung adenocarcinoma recurrence (Fig 5A), revealing significant associations between stage II (Odds Ratio [OR], 3.531; P < 0.001) and high POSTN expression (OR, 2.042; P < 0.001) with postoperative recurrence. Multivariate Cox regression analysis was then conducted, including factors with P-values < 0.05 from the univariate analysis (Fig 5B), which confirmed that stage II (OR, 3.444; P < 0.001) and high POSTN expression (OR, 2.022; P < 0.001) were independent risk factors for recurrence.

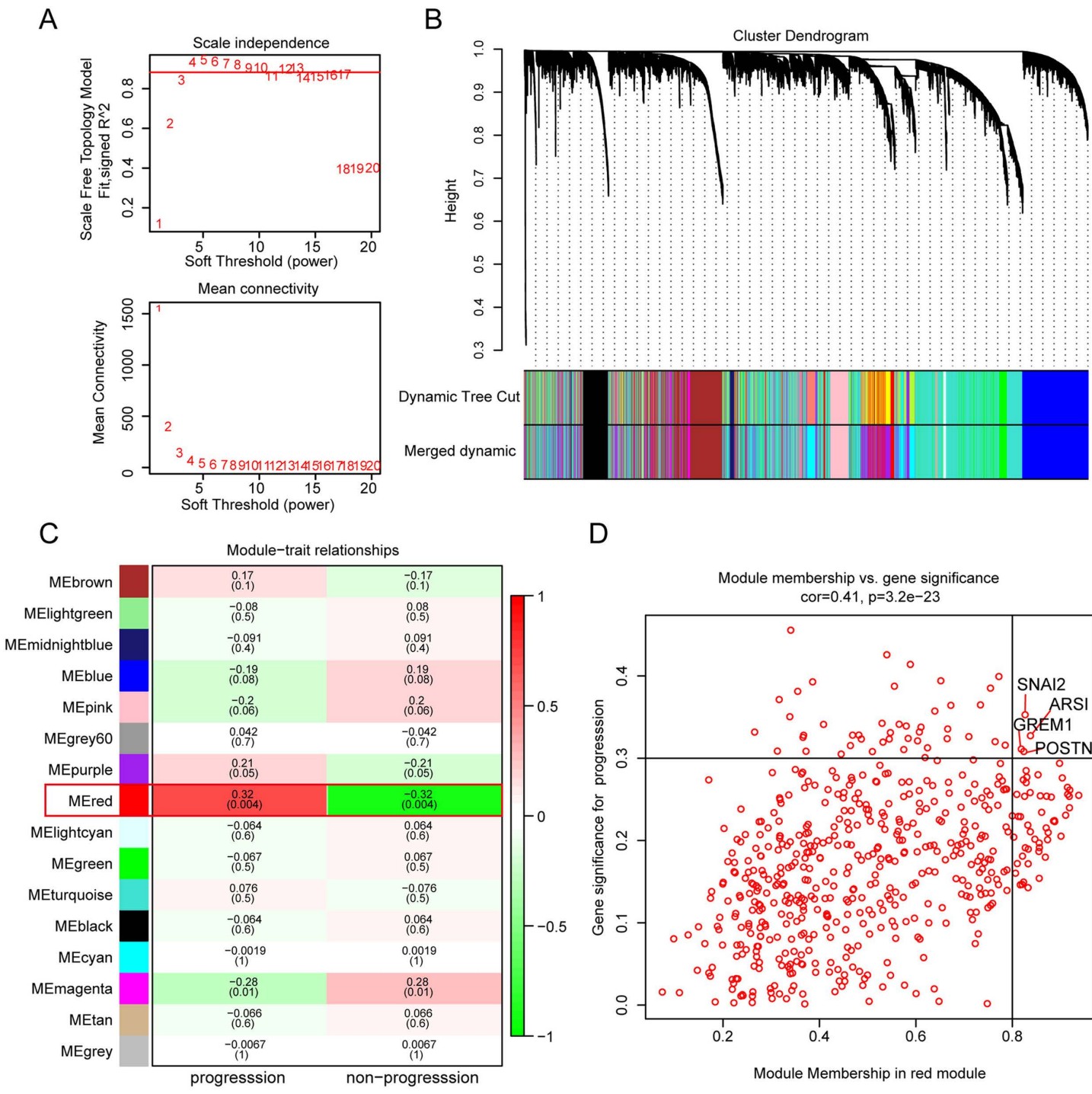

**Fig 2. Key gene acquisition and weighted gene co-expression network analysis (WGCNA) package to construct co-expression modules.**
(A-B) Analysis of threshold power and average connectivity of various soft-scale independence indicators. (C) Gene dendrogram clustering. Each branch in the graph represents a gene, and the colors under each indicate co-expression modules. (D) Heatmap of correlations between clinical traits, including recurrence and non-recurrence. Each column corresponds to a clinical trait and each row to a module. Each module contains the corresponding correlation coefficient and p-value. Green color indicates negative correlation and red color indicates positive correlation. (E) Correlations between MM and GS and clinical traits for the module of interest. Scatterplot of GS of disease process vs. MM for modules in red.

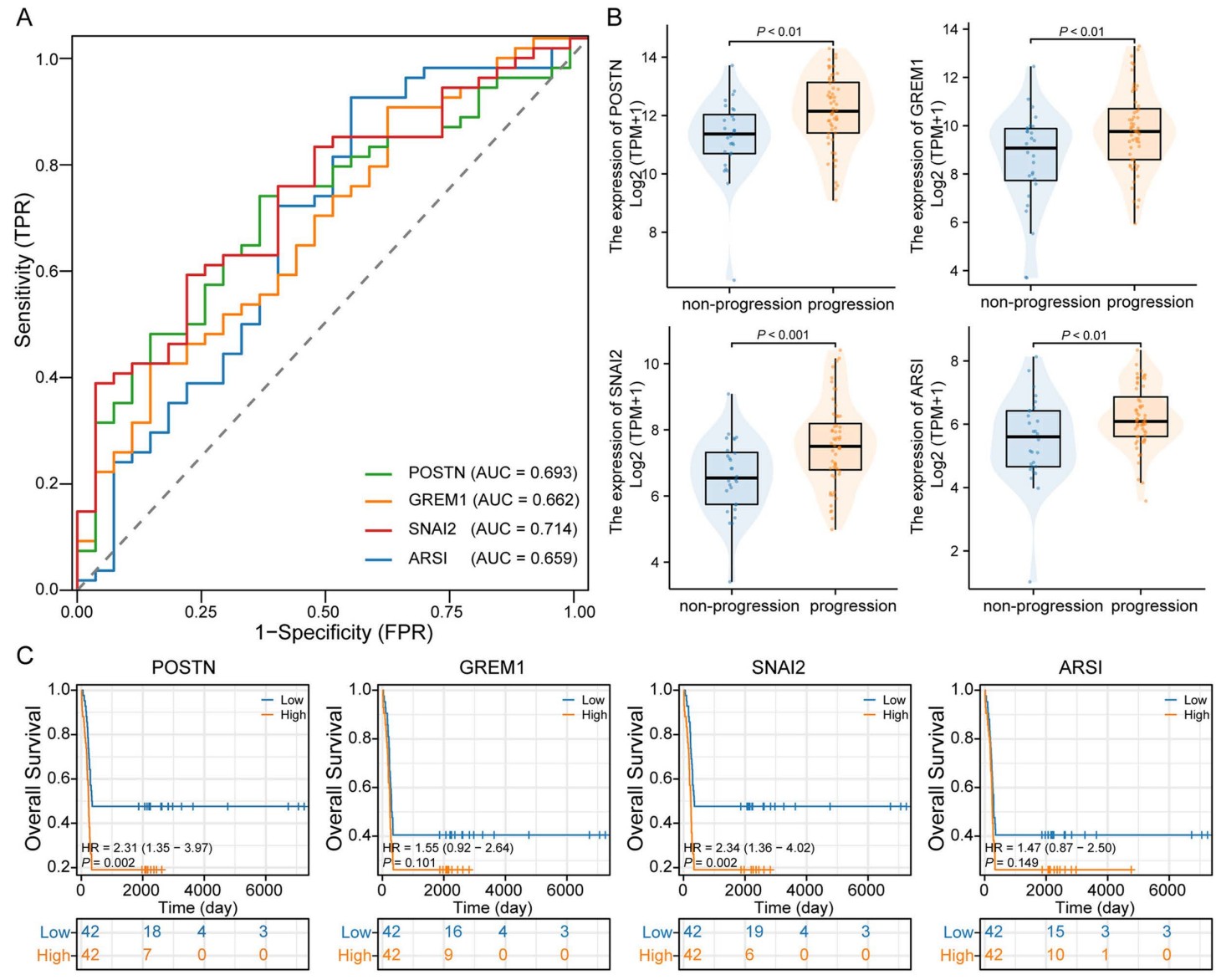

**Fig 3. Identification of key genes for disease progression.** (A) ROC curves of four genes in the TCGA database. (B) Expression differences of the four genes in the postoperative recurrence and non-recurrence groups of LUAD patients. (C) K-M curves showing patients' OS.

## 3.5 Subgroup analysis of key gene POSTN

To further assess the predictive generalization of POSTN, subgroup analysis was conducted (Fig 6A). Stratifying by gender, POSTN levels showed a positive correlation with recurrence risk, which was significant in both females (OR, 2.323; P < 0.001) and males (OR, 1.813; P < 0.01). Stratifying by age (<60 years or ≥60 years), the positive correlation between POSTN levels and recurrence risk persisted in both age groups (<60 years: OR, 2.023; 95% Confidence Interval (CI), 1.150–3.561; P < 0.05; ≥60 years: OR, 2.044; 95% CI, 1.388–3.010; P < 0.001). The expression of POSTN was significantly correlated with early-stage disease (In stage I patients, the multivariable-adjusted odds of recurrence increased

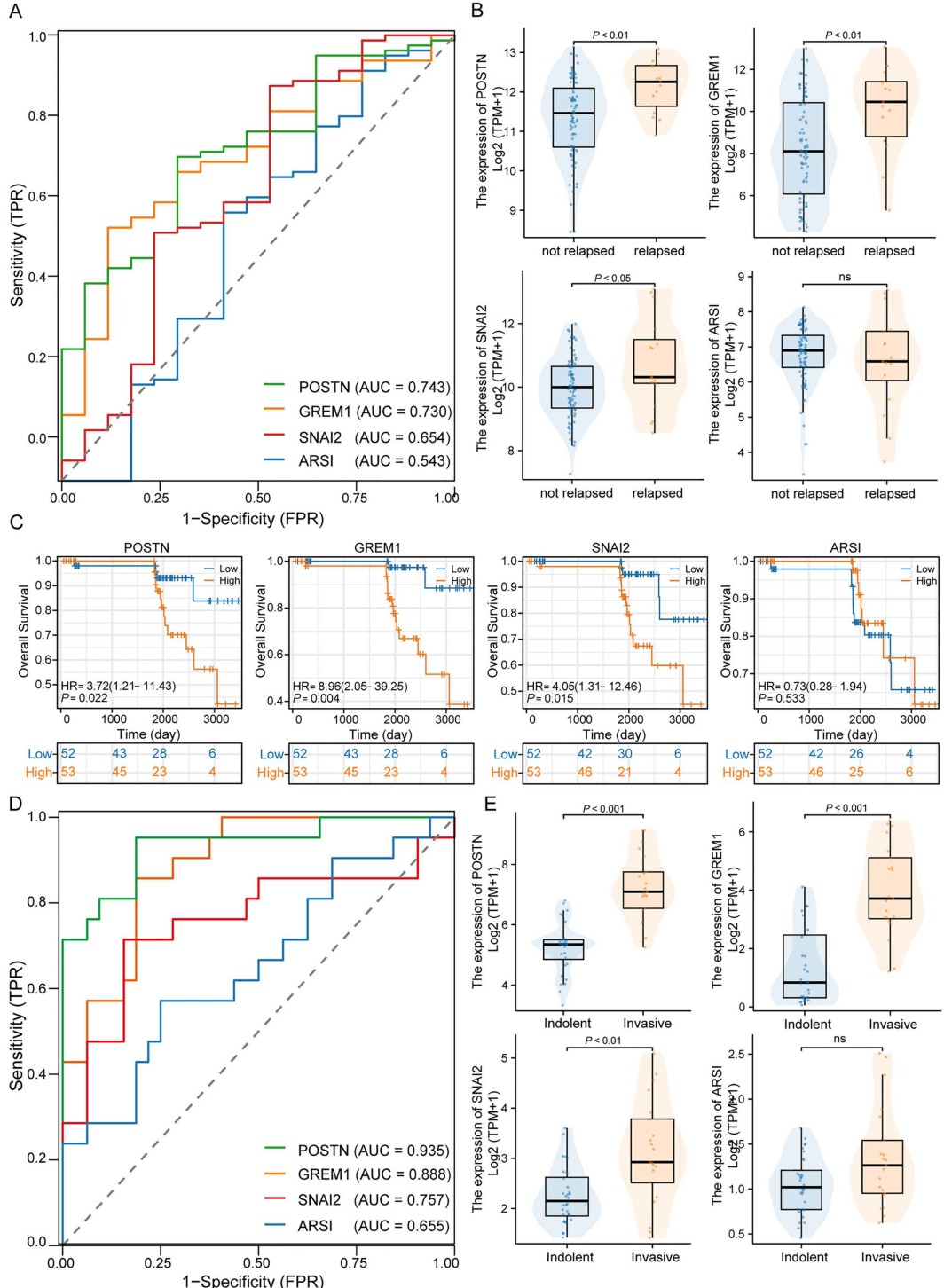

**Fig 4. External validation of POSTN in an independent dataset.** (A) ROC curves for four genes assessed for postoperative recurrence in LUAD patients in the GSE31210 dataset. (B) Differences between the four genes assessed in the GSE31210 dataset in the recurrence and non-recurrence groups. (C) Association between the four genes with different expression levels and OS in patients in the GSE31210 dataset. (D) ROC curves of the four genes in the GSE166722 dataset in assessing whether LUAD patient S is highly aggressive. (E) Differential expression of the four genes in the GSE166722 dataset in the aggressive and inert groups.

**Table 1. Description of the study population.**

| Outcome | Non-progression | Progression | Standardize difference | P-value |
|---|---|---|---|---|
| N | 116 | 73 | | |
| **POSTN, M±SD** | 11.313±1.095 | 12.126±1.146 | 0.726 (0.424, 1.027) | <0.001 |
| **Gender, N, (%)** | | | 0.065 (−0.227, 0.358) | 0.661 |
| Female | 61 (52.586) | 36 (49.315) | | |
| Male | 55 (47.414) | 37 (50.685) | | |
| **Age, N, (%)** | | | 0.093 (−0.200, 0.386) | 0.535 |
| <60 | 44 (37.931) | 31 (42.466) | | |
| ≥60 | 72 (62.069) | 42 (57.534) | | |
| **Pathological stage, N, (%)** | | | 0.599 (0.300, 0.898) | <0.001 |
| Stage I | 92 (79.310) | 38 (52.055) | | |
| Stage II | 24 (20.690) | 35 (47.945) | | |

Note: M±SD = Mean ± Standard Deviation; N = Number; % = Percentage.

2.25-fold per 1-unit increment in POSTN expression (OR = 2.252; 95% CI, 1.458–3.478; P<0.001), indicating a stronger predictive gradient compared with stage II (OR = 1.719).

Smooth curve fitting analysis demonstrated a linear relationship between POSTN and postoperative recurrence (Fig 6B). Additionally, stratified analysis, adjusting for potential confounding factors, confirmed the positive linear relationship between POSTN and recurrence in different genders, ages, and early-stage disease (Fig 6C).

### 3.6 Potential biological role of POSTN

To elucidate the potential biological mechanisms of POSTN in influencing disease recurrence, an interaction network was constructed to identify potential related interacted genes (Fig 7A). Through KEGG pathway analysis, POSTN was found to be associated with Extracellular Matrix (ECM)-receptor interaction and other related pathways (P<0.001), suggesting that POSTN likely affects cell invasion by influencing the extracellular matrix, thereby increasing the risk of postoperative recurrence (Fig 7B). Subsequently, gene enrichment analysis of POSTN was performed in the TCGA dataset (Fig 6C) and GSE31210 dataset (Fig 6D), which indicated that POSTN may primarily promote cancer progression by participating in the occurrence of epithelial-mesenchymal transition (EMT) (P<0.001).

### 3.7 Silencing POSTN inhibits proliferation, migration, and invasion of A549 and H1975 Cells

To further investigate the role of POSTN in cell migration and proliferation in vitro, siRNA targeting POSTN (siPOSTN-1, si-POSTN-2, and si-POSTN-3) were designed and synthesized. The results of the clone formation assay showed that the proliferation ability of both A549 and H1975 cells was inhibited after POSTN knockdown (Fig 8A). Additionally, the Transwell invasion analysis demonstrated that silencing POSTN significantly inhibited the invasion ability of A549 and H1975 cells (Fig 8B). Furthermore, the wound healing experiment showed that silencing POSTN significantly suppressed the migration of A549 and H1975 cells (Fig 8C). Overall, these results suggest that silencing POSTN can inhibit the proliferation, migration, and invasion of A549 and H1975 cells.

### 3.8 Overexpression of POSTN promotes proliferation, migration, and invasion of A549 and H1975 Cells

To further investigate the effects of POSTN overexpression on cell proliferation, migration, and invasion in vitro, we established POSTN-overexpressing A549 and H1975 cell models and conducted relevant experiments. The results of the colony formation assay showed that POSTN overexpression significantly promoted the proliferation of A549 and H1975 cells

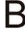

A

| Feature | N | HR(95% CI) | | Pvalue |
|---|---|---|---|---|
| Gender | | | | |
| Female | 97 (51.323%) | reference | | |
| Male | 92 (48.677%) | 1.175 (0.742, 1.859) | | 0.492 |
| Age | | | | |
| <60 | 75 (39.683%) | reference | | |
| ≥60 | 114(60.317%) | 1.025 (0.644, 1.631) | | 0.918 |
| Pathological stage | | | | |
| Stage I | 130(68.783%) | reference | | |
| Stage II | 59 (31.217%) | 3.420 (2.147, 5.446) | | <0.001 |
| POSTN | 11.627 ± 1.181 | 1.951 (1.518, 2.509) | | <0.001 |

B

| Feature | HR(95% CI) | | Pvalue |
|---|---|---|---|
| Pathological stage | | | |
| Stage I | reference | | |
| Stage II | 3.287 (2.062, 5.238) | | <0.001 |
| POSTN | 1.945 (1.506, 2.510) | | <0.001 |

**Fig 5. Univariate and multivariate Cox analysis by sex, age, stage, and POSTN.** (A) Univariate Cox forest plot of disease recurrence versus gender, age, stage, and POSTN (B) Multivariate Cox forest plot of disease recurrence versus stage and POSTN.

(Fig 9A). In the Transwell migration assay, the number of migratory cells in the overexpression group was significantly higher than that in the OE_NC group (Fig 9B). The wound healing assay indicated that the wound healing rate of the OE_POSTN group was significantly faster than that of the OE_NC group at both 24 and 48 hours (Fig 9C). In summary, POSTN overexpression significantly enhances the proliferation, migration, and invasion of A549 and H1975 cells, which is contrary to the inhibitory effects observed after POSTN silencing, further confirming the crucial role of POSTN in cellular biological behavior.

## 4 Discussion

Our study identified POSTN as an independent risk factor for recurrence in early-stage LUAD, with consistent predictive performance across multiple cohorts. The mechanism underlying POSTN-mediated recurrence may involve its regulation of epithelial-mesenchymal transition (EMT), a key process driving tumor invasion and metastasis.

POSTN is a multifunctional extracellular matrix protein involved in various physiological processes, including bone regeneration, cardiac remodeling, skin response to injury, and renal development [19]. Initial studies have identified

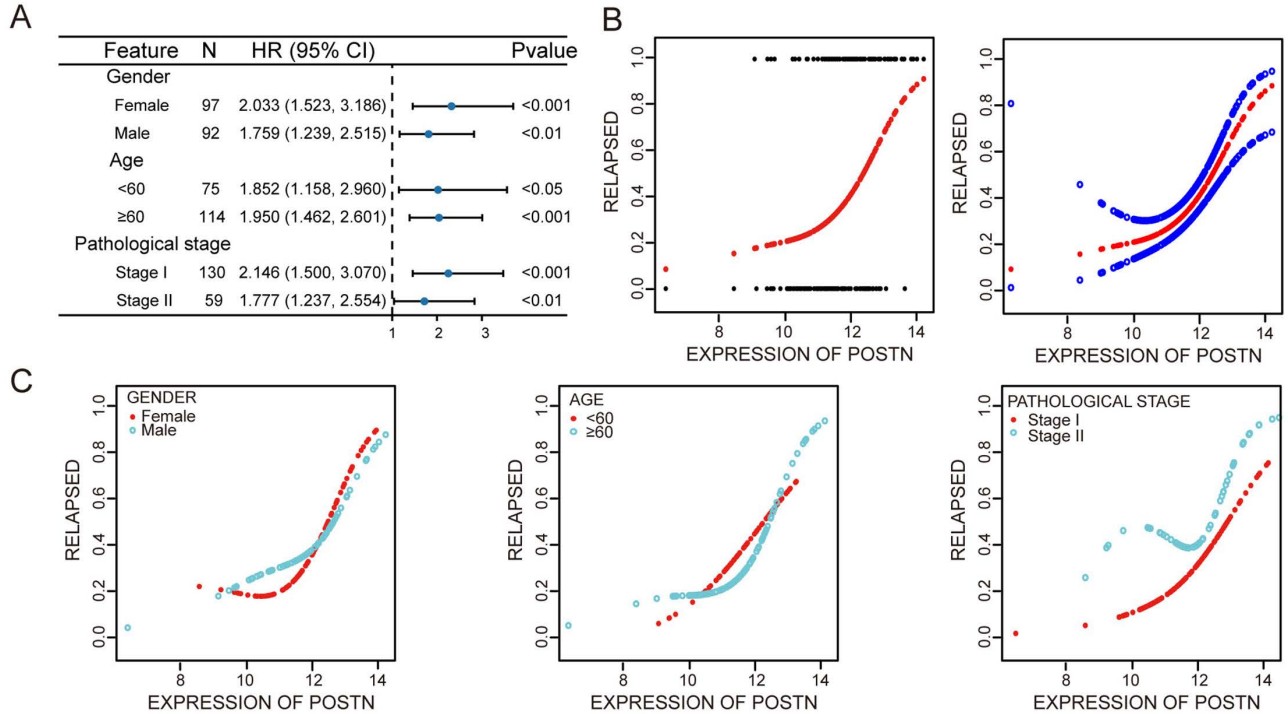

**Fig 6. Stratified analysis of POSTN and relationship with disease progression.** (A) Forest plot of stratified analysis (B) Relationship between POSTN levels and disease progression outcomes. Each black dot represents one sample. Vertical coordinates 0.0 indicates no recurrence, 1.0 indicates recurrence, and solid lines indicate the corresponding LMR distribution for each sample. The solid line on the right indicates a smooth curve fit between the variables, and the blue bar indicates the 95% confidence interval of the fit. (C) Relationship between POSTN and recurrence, stratified by clinical characteristics. Stratified by gender, age, and type of staging, respectively.

POSTN as a specific factor for osteoblasts, and its upregulation has been observed in various human tumors [20,21]. In addition, POSTN acts as a scaffold for many other proteins and is involved in signal transduction from cells to the matrix and epithelial-to-mesenchymal transition, thereby promoting cancer progression [22] and the development of chemoresistance [23]. Emerging evidence supports that POSTN promotes EMT to enhance the motility of tumor cells [24,25]. In lung adenocarcinoma cell lines (A549 and H1975), POSTN silencing upregulates the epithelial marker E-cadherin and downregulates the mesenchymal markers N-cadherin and Vimentin; conversely, POSTN overexpression exerts the opposite effect [26]. This regulatory pattern is consistent with our in vitro findings, where POSTN silencing inhibits cell invasion and migration, while overexpression enhances these phenotypes.

It is worth noting that POSTN is both an intracellular signaling molecule and a matrix protein that can be secreted extracellularly. This study found that the intracellular level of POSTN directly regulates the proliferation and invasion of tumor cells, exhibiting a cell-autonomous effect. However, existing studies have shown that soluble POSTN secreted by CAFs in the tumor microenvironment can paracrinally activate adjacent tumor cells through integrin αVβ3, driving EMT and promoting metastasis [19]. Even if POSTN in the tumor cells themselves is knocked down, POSTN from non-cell-autonomous sources may still maintain residual invasive ability, which may be one of the reasons for early recurrence in some patients with low POSTN expression.

Notably, our WGCNA analysis identified SNAI2 as another key gene associated with LUAD recurrence, and correlation analysis showed a high co-expression between POSTN and SNAI2 (R = 0.74, P = 1.6e-15). SNAI2 is a typical EMT transcription factor that can inhibit the expression of E-cadherin, thereby driving mesenchymal transformation [27,28],

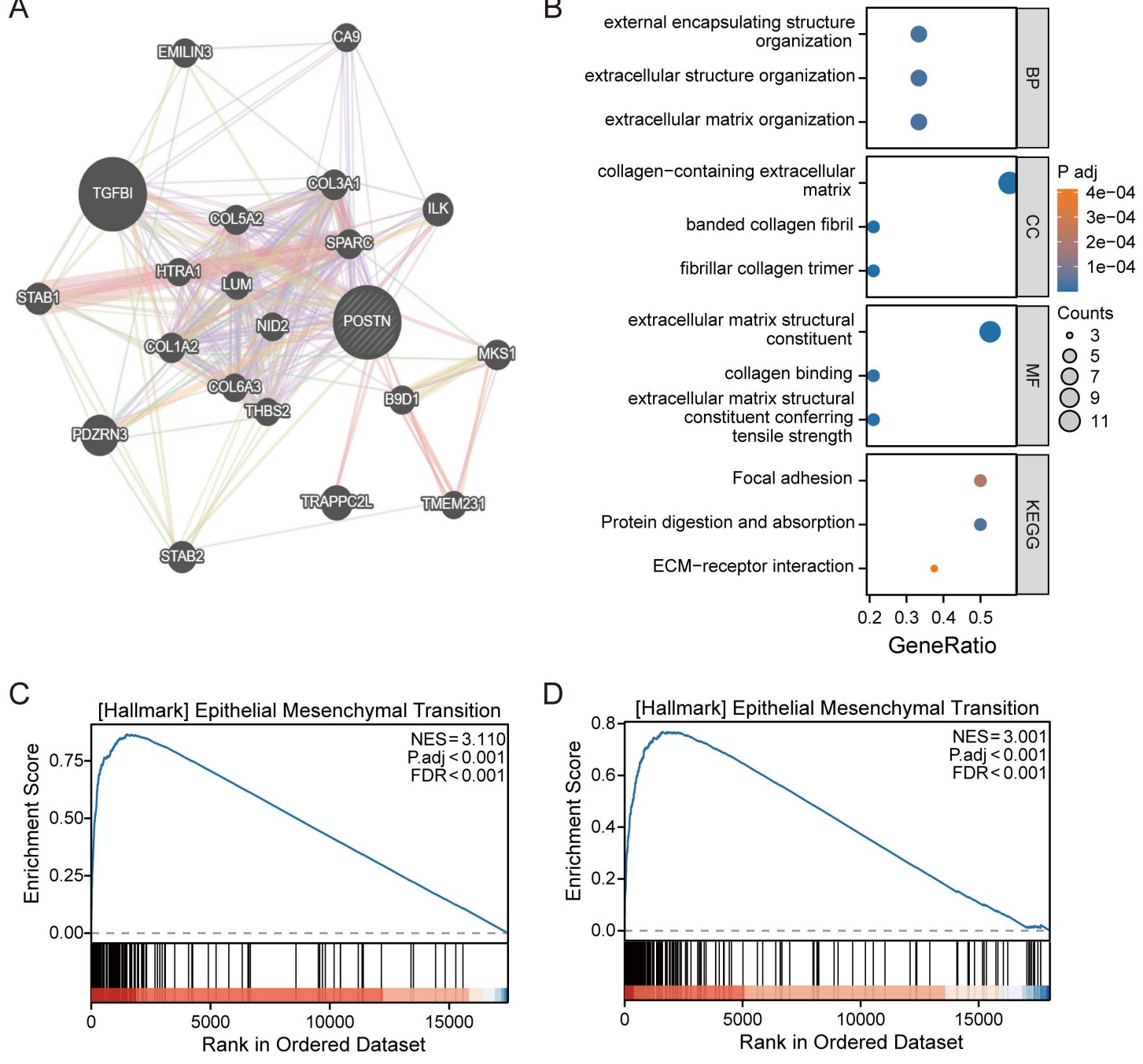

**Fig 7. Potential mechanisms by which POSTN affects recurrence.** (A) Interaction network diagram of POSTN (B) Bubble diagram of gene associated with the target pathway (C) (D) Gene enrichment analysis in TCGA and GSE31210 datasets.

suggesting that it may synergize with POSTN in EMT regulation. However, direct experimental evidence for their interaction is lacking. Future studies will address this by performing combined knockdown/overexpression experiments and molecular interaction assays (co-immunoprecipitation) to clarify their regulatory hierarchy in the EMT pathway, which may lead to the discovery of new therapeutic targets for combined intervention.

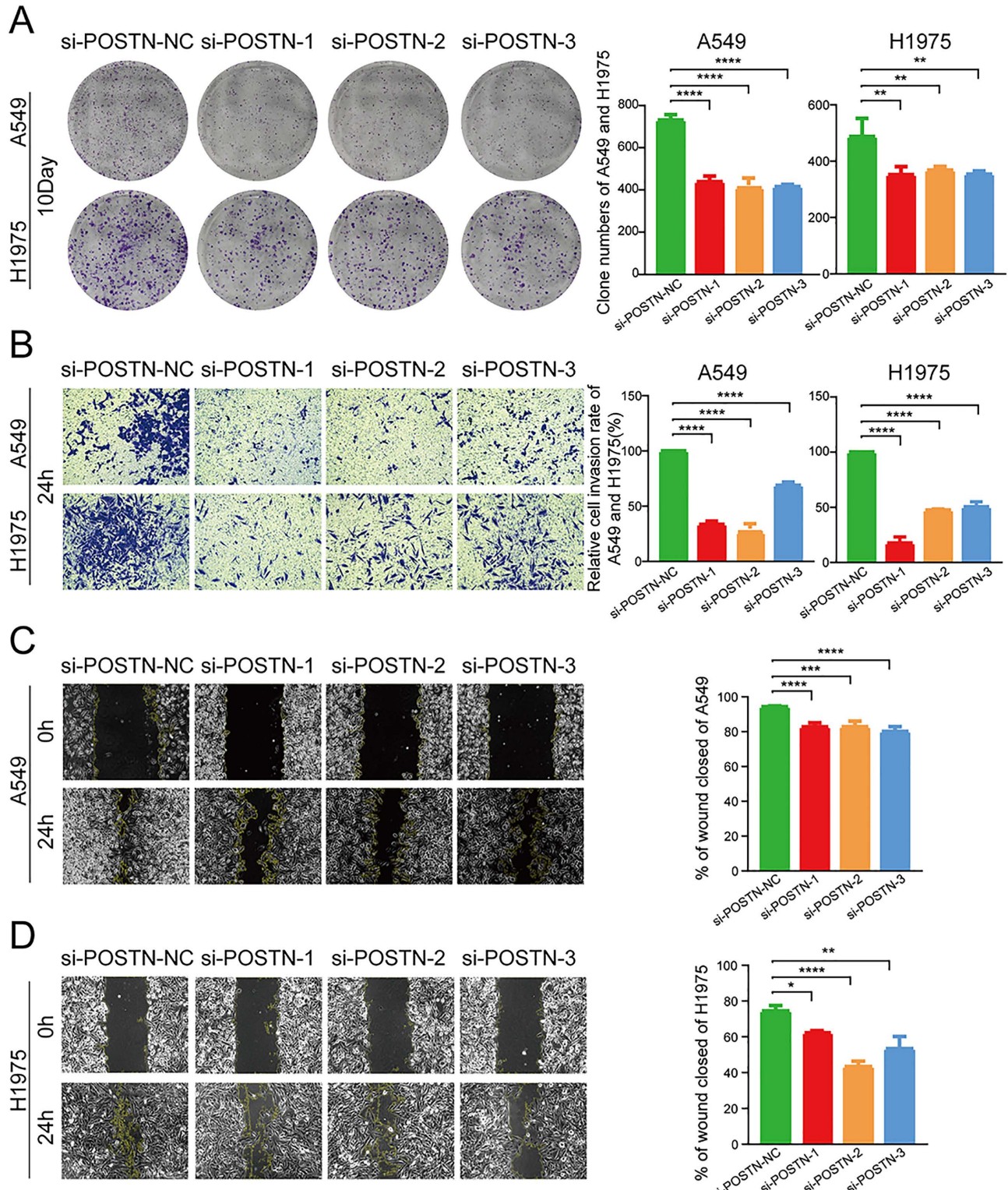

**Fig 8. Effect of POSTN on proliferation, migration invasion of A549 and H1975 cells.** (A) Clone formation assay in the silenced and control groups. (B) Transwell invasion assay for analyzing the invasive capacity of A549 and H1975 cells in the silent and control groups. (C) Wound healing assay for silent and control groups to detect the migration ability of A549 and H1975 cells. *P<0.05; **P<0.01; ***P<0.001; ****P<0.001.

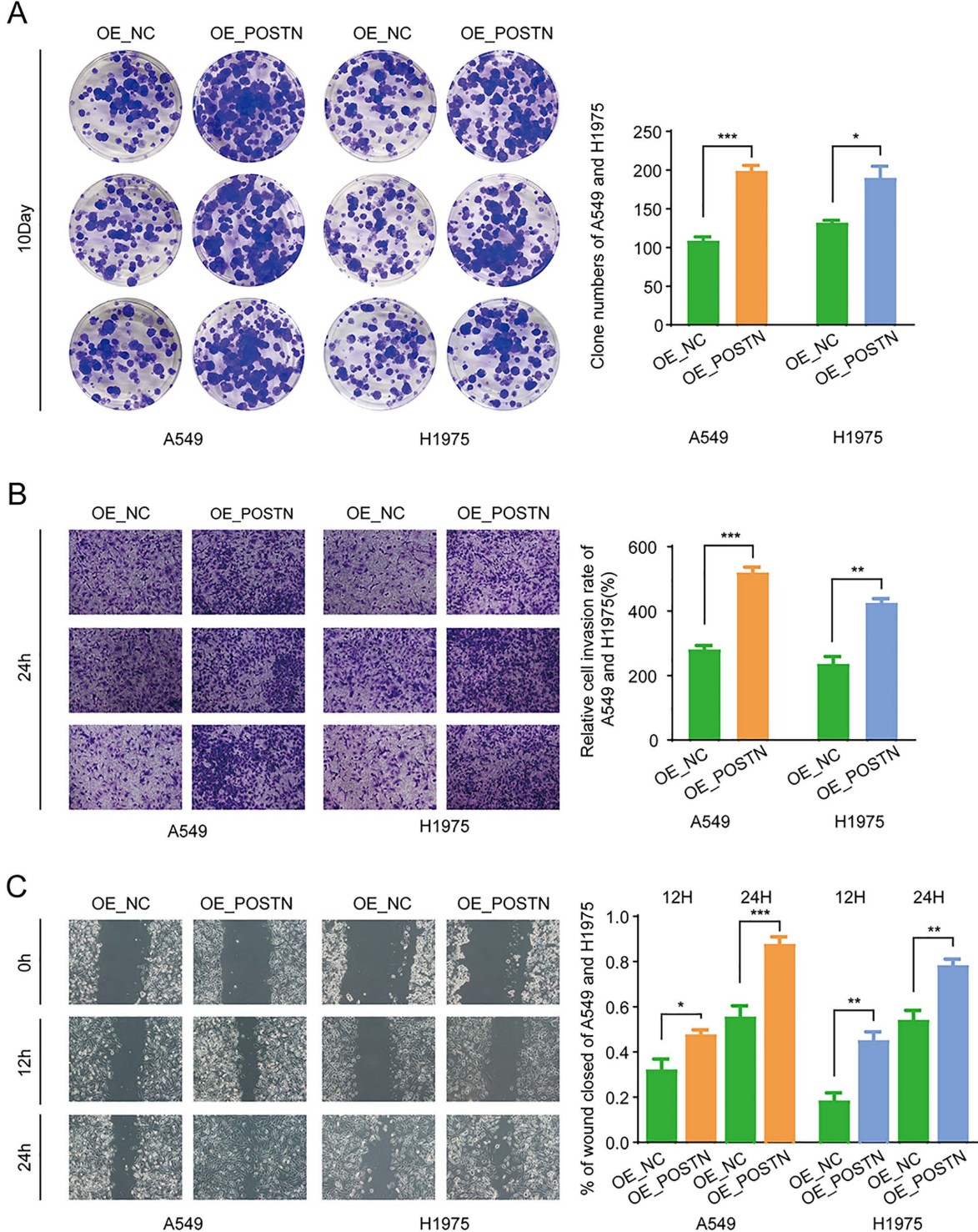

**Fig 9. Effects of POSTN Overexpression on Proliferation, Migration, and Invasion of LUAD Cells.** (A) Results and statistical analysis of clone formation assays comparing overexpression and control groups. (B) Transwell assays to assess cell invasion potential, with invasive cell counts. (C) Wound healing assays to measure cell migration activity and calculate migration rates. Significance levels are indicated as *P<0.05; **P<0.01; ***P<0.001; ****P<0.001.

To further explore the regulatory basis of recurrence-related genes, we analyzed the upstream transcription factors of the four key genes (SNAI2, ARSI, GREM1, POSTN) using the hTFtarget database, FIMO_JASPAR tool, and ENCODE database. Intersection analysis revealed no common master transcription factor, indicating that these genes may participate in LUAD recurrence through independent upstream pathways. This finding suggests that the pro-recurrence function of POSTN may be regulated by a unique set of upstream factors. Combining multi-database predictions with experimental validation (such as ChIP-qPCR) to identify POSTN-specific upstream regulators may further discover emerging regulatory networks for recurrence risk in lung cancer patients.

A critical consideration for prognostic biomarkers is their generalizability across clinical subtypes. Our supplementary analysis of the TCGA-LUAD and GSE31210 datasets showed no significant differences in POSTN expression between EGFR mutant and wild-type patients, between smokers and non-smokers, or between genders (S1 Fig). This indicates that the predictive value of POSTN is not limited to specific molecular subtypes or clinical characteristics, supporting its potential as a "subtype-independent" biomarker. This generalizability enhances its clinical utility, as it can identify high-risk recurrence populations among diverse patient groups.

This study has several limitations. First, the sample size is still relatively small, and larger multi-center cohorts are needed to validate the prognostic value of POSTN. Second, the synergistic mechanism between POSTN and SNAI2 in EMT requires further experimental exploration. Third, the upstream regulatory network of POSTN and other key genes deserves in-depth study to identify potential master regulators. Fourth, although our in vitro experiments support the cell-autonomous effect of POSTN, the non-autonomous role of POSTN in LUAD recurrence (paracrine effects from cancer-associated fibroblasts) remains to be clarified through co-culture or conditioned medium assays.

## 5 Conclusion

Elevated POSTN expression may promote cell migration by affecting epithelial-to-mesenchymal transition, increasing the risk of recurrence or progression in early-stage LUAD patients following treatment. POSTN can serve as a biomarker for monitoring treatment response.

## Supporting information

**S1 Fig. Expression of POSTN in different subtypes.** (A) The distribution of POSTN in different variable populations in the TCGA database. (B) The distribution of POSTN in different variable populations in the GEO dataset.
(TIF)

**S1 Table. Abbreviations and full names of words.**
(DOCX)

**S2 Table. Data inclusion and analysis purpose.**
(DOCX)

**S1 File. GSE31210.**
(XLSX)

**S2 File. GSE166722.**
(XLSX)

**S3 File. TCGA&WGCNA.**
(XLSX)

## Author contributions

**Conceptualization:** Juan Hu, Dan Wang.

**Data curation:** Youde Xiao, Xuelian Lin, Hui Gong.

**Formal analysis:** Youde Xiao, Juan Hu.

**Investigation:** Xuelian Lin, Hui Gong.

**Methodology:** Xuelian Lin, Hui Gong.

**Resources:** Juan Hu.

**Software:** Youde Xiao, Juan Hu.

**Supervision:** Juan Hu, Dan Wang.

**Writing – original draft:** Youde Xiao, Xuelian Lin, Hui Gong.

**Writing – review & editing:** Youde Xiao.

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
