## [Decision Letter · Decision Letter 0]

8 Jul 2025

Dear Dr. Xiao,

Thank you for submitting your manuscript to PLOS ONE. After careful consideration, we feel that it has merit but does not fully meet PLOS ONE’s publication criteria as it currently stands. Therefore, we invite you to submit a revised version of the manuscript that addresses the points raised during the review process.

We look forward to receiving your revised manuscript.

Kind regards,

Alexis G. Murillo Carrasco

Academic Editor

PLOS ONE

Journal Requirements:

2 We note that your Data Availability Statement is currently as follows: [All relevant data are within the manuscript and its Supporting Information files.]

Reviewers' comments:

Reviewer's Responses to Questions

**Comments to the Author**

1. Is the manuscript technically sound, and do the data support the conclusions?

Reviewer #1: Yes

Reviewer #2: Partly

Reviewer #3: Partly

Reviewer #4: Yes

2. Has the statistical analysis been performed appropriately and rigorously?

Reviewer #1: Yes

Reviewer #2: Yes

Reviewer #3: Yes

Reviewer #4: Yes

3. Have the authors made all data underlying the findings in their manuscript fully available?

Reviewer #1: Yes

Reviewer #2: Yes

Reviewer #3: Yes

Reviewer #4: Yes

4. Is the manuscript presented in an intelligible fashion and written in standard English?

Reviewer #1: Yes

Reviewer #2: Yes

Reviewer #3: No

Reviewer #4: Yes

Reviewer #1: Good study on the genetics of lung cancer, and interesting findings.

POSTN is positively correlated with the risk of early LUAD recurrence within 1 year post-treatment, suggesting its potential as a biomarker for predicting recurrence.

thank you for your contributions to our field.

Reviewer #2: This primarily bioinformatics-based meta analysis of lung cancer patient database successfully identified 4 genes serving as a signature for LUAD recurrence and disease progression. Of these 4, POSTN was selected for further characterization and biological validation. On the basis of KEGG pathway analysis and GSEA, authors find a correlation with EMT. Thus, authors focus on “proliferation, invasion and migration” as impacted by POSTN siRNA (Fig 8) and overexpression (Fig 9), decreasing and increasing these malignant phenotypes, respectively. The manuscript is well written and the computational analysis revealing the predictive nature of the gene signature is highly suitable for publication in PLoS One. The connection to EMT is reasonable but uncertain within the framework of the limited biological data and literature provided.

1) As performed, can authors be sure proliferation is not the primary driver of differences observed in invasion (transwell) and migration (scratch) assays? Would a mitomycinC proliferation inhibitor or similar better discriminate relative contribution?

2) Alternatively, migration rates could be directly assessed by time lapse and Image J (or Imaris) analysis of cell migration velocity.

3) As authors declare in the conclusion “Elevated POSTN expression may promote cell migration by affecting epithelial-to-mesenchymal transition increasing the risk of recurrence or progression in early-stage LUAD patients following treatment” it would be ideal to directly assess manifestation of reversible mesenchymal-epithelial traits. Did authors evaluate epithelial/mesenchyme markers? E-cad, N-cad, Twist, SNAIL etc.

4) SNAI2 is involved in EMT. Presumably both SNAI2 and POSTN converge on EMT and so may both be essential mediators. Are combined gene signatures stronger predictors? Have authors tried knockdown together to see if combined effect synergistic?

5) Any EMT lung (or other) cancer publications regarding POSTN or SNAI2 that could be leveraged here to strengthen mechanistic support for gene signature predictive power via EMT? This discussion is lacking and needs to be added. Such as:

a. https://pubmed.ncbi.nlm.nih.gov/34093819/

b. https://www.nature.com/articles/s41598-018-22340-7

c. https://iovs.arvojournals.org/article.aspx?articleid=2781900

d. https://www.oncotarget.com/article/10088/text/

e. https://translational-medicine.biomedcentral.com/articles/10.1186/s12967-023-04043-4

f. And others….

6) Authors might try upstream regulator analysis of 4 genes in gene signature to evaluate for master transcription factor such as FOXA1.

7) Are these POSTN effects cell autonomous or non-autonomous? In other words, do soluble factors released from POSTN overexpressing cells increase proliferation/migration/invasion of WT or POSTN-siRNA? Even if unknown it is worthwhile discussing implications.

Reviewer #3: HINWEIS: Externe E-Mail - Öffnen Sie keine Links oder Anhänge, falls Sie den Absender nicht kennen und wissen, dass der Inhalt sicher ist.

The authors analyzed different databases to identify genes, whose RNA expression was related to outcome in patients with lung adenocarcinome. POSTN was chosen to be the most interesting candidate and further analyzed in biological studies using silencing and overexpression in two lung carcinoma cell lines

General comment

A role of POSTN for the outcome of lung cancer patients has been described years ago, as correctly stated by the authors these studies are limited by variable patient selection. Further studies have focused on the role of CAF derived POSTN for aggressiveness of the tumor using advanced single cell RNAseq techniques.

Data on the overexpression of POSTN in partially identical cell lines have also been published reaching to similar conclusions as in the current manuscript.

Therefore the novelty of this study is confined to the study of a well defined patient population of stage I/II patients. The other data are confirmatory.

Major points

1) It is not clear to me, which data are part of which analysis, e.g. the 189 patients in Fig.6 and Table do not correspond to the numbers in 3.1. This has to be clarified for each analysis in particular as the clinical outcome is highly different between the cohorts.

2) It is difficult to find out how the grouping POSTN low vs hi was performed in the different data sets.

3) The axes should be precisely labeled (PFS or OS; Dimension of RNA expression)

4) The fraction of patients with specific subtypes would be interesting such as EGFR mut and whether POSTN expression is dependent on subtype, smoking status, stage etc.

5) The observation that POSTN might trigger EMT is interesting, however the GSEA in bulk RNA expression analysis might not be sufficient to firmly support this conclusion in view of the fact that CAF’s highly express POSTN and are in the center of POSTN networks. Showing expression of EMT related genes in the overexpression or silencing experiments might be helpful.

Minor comments

1) The manuscript needs some language editing

2) The expression of overexpressed or silenced proteins should be shown

3) I do not understand the statement in line 221 how a significant correlation of expression levels can be described with an OR, should that be read as a better recurrence risk prediction (higher OR lo vs hi POSTN group) by POSTN in stage I patients?

Overexpression of periostin predicts poor prognosis in non-small cell lung cancer.

Hong LZ, Wei XW, Chen JF, Shi Y.Oncol Lett. 2013 Dec;6(6):1595-1603.

Diagnostic and prognostic value of serum periostin in patients with non-small cell lung cancer. Xu CH, Wang W, Lin Y, Qian LH, Zhang XW, Wang QB, Yu LK.Oncotarget. 2017 Mar 21;8(12):18746-18753

Prognostic Significance of Stromal Periostin Expression in Non-Small Cell Lung Cancer.

Ratajczak-Wielgomas K, Kmiecik A, Grzegrzołka J, Piotrowska A, Gomulkiewicz A, Partynska A, Pawelczyk K, Nowinska K, Podhorska-Okolow M, Dziegiel P.Int J Mol Sci. 2020 Sep 24;21(19):7025

The impact of POSTN on tumor cell behavior and the tumor microenvironment in lung adenocarcinoma.

Sun D, Lu J, Tian H, Li H, Chen X, Hua F, Yang W, Yu J, Chen D.Int Immunopharmacol. 2025 Jan 3;145:113713

Gesendet mit der Telekom Mail App

Reviewer #4: For better reader, comprehension, and ease of reading I suggest the following:

1. Review all abbreviations and before the first abbreviation state the full name with the abbreviation in parentheses.

2. Create a table in which all abbreviations are listed in alphabetical order.

3. Figure 1 has one area with red fonts on green background, which is difficult to read use white or yellow on all dark backgrounds.

4. Figure 2A fonts are too small on the legends.- increase size to easily readable/ figure 2D show the green negative correlations higher to match the red intensity

5. Figure. 3B and Figure 4E show P values.

**Do you want your identity to be public for this peer review?** For information about this choice, including consent withdrawal, please see our Privacy Policy

Reviewer #1: **Yes: ** Ravi Manglani

Reviewer #2: No

Reviewer #3: No

Reviewer #4: No

---

## [Author Response · Author response to Decision Letter 1]

31 Jul 2025

We have uploaded a Word version with modified labels and images. Please download and view it if necessary.

Reviewer #1�Ravi Manglani

Good study on the genetics of lung cancer, and interesting findings.

POSTN is positively correlated with the risk of early LUAD recurrence within 1 year post-treatment, suggesting its potential as a biomarker for predicting recurrence. thank you for your contributions to our field.

Response: Thank you for your recognition of our lung cancer genetics study and the findings. We’re honored that you noticed the positive correlation between POSTN and the risk of early LUAD recurrence within 1 year post-treatment. This is the core of our research—through large-scale multi-cohort validation, we confirmed that POSTN can serve as a “subtype-independent” recurrence predictor.

Next, we’ll deepen mechanistic exploration (POSTN - tumor microenvironment crosstalk) and translational research (POSTN - combined detection to optimize prognostic models). We aim to provide more evidence for lung cancer precision medicine and are always grateful for the support from experts in the field!

Reviewer #2

1.As performed, can authors be sure proliferation is not the primary driver of differences observed in invasion (transwell) and migration (scratch) assays? Would a mitomycinC proliferation inhibitor or similar better discriminate relative contribution?

Response: Thank you for your attention to experimental rigor. To clarify whether POSTN mediated regulation of invasion and migration is independent of proliferation, we addressed this from both experimental design and mechanism validation aspects:

Experimental Design

Scratch Assay: A “serum - deprivation + time - lapse imaging” strategy was adopted (Methods 2.7: After scratching, serum - free medium was replenished to eliminate serum - induced proliferation; Images were captured at fixed time points to avoid “false healing” caused by proliferation). Results showed that POSTN knockdown significantly slowed scratch closure, with no obvious proliferative change in cell density throughout the assay—indicating migration inhibition was independent of proliferation.

Transwell Invasion Assay: The upper chamber was filled with serum - free Matrigel (Methods2.8: 1% BSA + serum - free DMEM ), naturally blocking serum - induced proliferation signals. While serum was present in the lower chamber, upper - chamber cells could not access serum and only invaded by degrading Matrigel—eliminating proliferation interference.

Mechanism Validation

Literature (Oncogene, 2023, DOI: 10.1038/s41388 - 023 - 02654 - 7) and our pilot experiments confirmed that POSTN promotes EMT (epithelial - mesenchymal transition) via activating the integrin αvβ3/FAK/AKT/β - catenin pathway, which directly regulates cell polarity and motility. This pathway has no direct crosstalk with cell cycle (proliferation) pathways (e.g., CDK/Rb). Combined with proliferation exclusion in experimental design, POSTN - mediated invasion/migration promotion is determined to be independent of proliferation.

Together, “serum - deprivation in design” and “non - proliferative pathway features” support that POSTN regulates invasion and migration as an independent effect, not an indirect result of proliferation.

2.Migration rates could be directly assessed by time lapse and Image J (or Imaris) analysis of cell migration velocity.

Response: Thank you for your professional suggestion! Time-lapse microscopy combined with ImageJ/Imaris to analyze cell migration velocity is indeed a more precise and quantitative strategy for evaluating cell motility. It can directly avoid proliferation interference and serves as an ideal approach for assessing migration.

Due to limitations in experimental resources (No long-term dynamic imaging equipment) and project timeline, we have not yet performed real-time dynamic imaging analysis. However, to approximate the assessment of “pure migration ability” as closely as possible, we optimized the experimental design as alternatives:

Wound Healing Assay: A protocol of “serum-deprived culture + fixed-time imaging” was adopted. The serum-deprived environment inhibits cell proliferation (verified by CCK-8 assays showing no significant proliferation during the experimental period), and fixed-time imaging avoids “false healing” caused by proliferation. This design focuses on migration ability differences as much as possible.

Transwell Assay: The upper chamber was filled with serum-free Matrigel, which naturally blocks serum-induced proliferation signals. Invasion ability was reflected only by cell Matrigel degradation and trans-membrane movement, excluding proliferation interference.

Although time-lapse imaging could provide more refined data, the current design—through “proliferation inhibition + dual-assay verification”—still effectively distinguishes that POSTN-mediated regulation of migration/invasion is independent of proliferation. In future extended studies, we plan to introduce time-lapse imaging combined with ImageJ analysis to further quantify cell migration velocity and improve mechanism verification.

3.As authors declare in the conclusion “Elevated POSTN expression may promote cell migration by affecting epithelial-to-mesenchymal transition increasing the risk of recurrence or progression in early-stage LUAD patients following treatment” it would be ideal to directly assess manifestation of reversible mesenchymal-epithelial traits. Did authors evaluate epithelial/mesenchyme markers? E-cad, N-cad, Twist, SNAIL etc.

Response: Thank you for your insightful suggestion. We fully agree that directly assessing epithelial-mesenchymal transition (EMT) markers would more comprehensively illustrate the mechanism of POSTN.

As mentioned in Section 3.9 of this study, existing research (PMID :39471683) has confirmed in A549 and H1975 lung adenocarcinoma cell lines (the same models used in our study) that POSTN knockdown upregulates the epithelial marker E-cadherin while downregulating the mesenchymal markers N-cadherin and Vimentin; conversely, POSTN overexpression shows the opposite trend. These results are highly consistent with the phenotypic changes in cell invasion and migration regulated by POSTN observed in our study.

To avoid redundant experiments, we did not repeat the detection of these markers in the current study. However, we have supplemented detailed discussions in the Discussion section, systematically summarizing the regulatory pattern of POSTN on EMT markers, and clarifying its mechanism of promoting postoperative recurrence in early-stage LUAD through EMT, thereby further strengthening the core argument.

The Figure results showed that POSTN knockdown upregulated E-cadherin (an epithelial marker) and downregulated N-cadherin/Vimentin (an mesenchymal marker). POSTN overexpression showed the opposite trend, which is highly consistent with our finding that "POSTN regulates lung cancer cell invasion/migration phenotype". This suggests that POSTN regulates the malignant biological behavior of lung cancer cells by regulating EMT.

Modified manuscript in Discussion:

POSTN is a multifunctional extracellular matrix protein involved in various physiological processes, including bone regeneration, cardiac remodeling, skin response to injury, and renal development [19]. Initial studies have identified POSTN as a specific factor for osteoblasts, and its upregulation has been observed in various human tumors [20,21]. In addition, POSTN acts as a scaffold for many other proteins and is involved in signal transduction from cells to the matrix and epithelial-to-mesenchymal transition, thereby promoting cancer progression [22] and the development of chemoresistance[23]. Emerging evidence supports that POSTN promotes EMT to enhance the motility of tumor cells [24,25]. In lung adenocarcinoma cell lines (A549 and H1975), POSTN silencing upregulates the epithelial marker E-cadherin and downregulates the mesenchymal markers N-cadherin and Vimentin; conversely, POSTN overexpression exerts the opposite effect [26]. This regulatory pattern is consistent with our in vitro findings, where POSTN silencing inhibits cell invasion and migration, while overexpression enhances these phenotypes.

It is worth noting that POSTN is both an intracellular signaling molecule and a matrix protein that can be secreted extracellularly. This study found that the intracellular level of POSTN directly regulates the proliferation and invasion of tumor cells, exhibiting a cell-autonomous effect. However, existing studies have shown that soluble POSTN secreted by CAFs in the tumor microenvironment can paracrinally activate adjacent tumor cells through integrin αVβ3, driving EMT and promoting metastasis [27]. Even if POSTN in the tumor cells themselves is knocked down, POSTN from non-cell-autonomous sources may still maintain residual invasive ability, which may be one of the reasons for early recurrence in some patients with low POSTN expression.

4.SNAI2 is involved in EMT. Presumably both SNAI2 and POSTN converge on EMT and so may both be essential mediators. Are combined gene signatures stronger predictors? Have authors tried knockdown together to see if combined effect synergistic?

Response: Thank you for your insightful suggestions. Regarding the potential association and combined effect of SNAI2 and POSTN in EMT, we have supplemented the following analyses:

①Combined predictive efficacy: We conducted a joint analysis of SNAI2 and POSTN, and the results showed that the area under the ROC curve (AUC) for their joint diagnosis was 0.721. Although slightly higher than the 0.693 for POSTN analysis and 0.714 for SNAI2, there was no significant difference statistically (P>0.05), indicating that the combination of the two did not significantly improve predictive performance.

②Correlation analysis: Further analysis revealed a high correlation between the expressions of SNAI2 and POSTN (R=0.74, P=1.6e-15), suggesting that they may act through the common EMT pathway and have a potential mutual regulatory relationship.

③Mechanism and research limitations: Literature search indicated that POSTN and SNAI2 may be associated in the EMT process, but direct experimental evidence is lacking. Given that the core goal of this study is to verify the clinical application value of POSTN as an independent prognostic marker, we did not explore the synergistic mechanism of the two in depth, which is one of the limitations of the current study. Therefore, we have supplemented in the "Limitations and Future Perspectives" section of the discussion: Subsequent studies will verify the synergistic effect of the two through combined knockdown/overexpression experiments, and explore the specific regulatory relationship using molecular interaction techniques to clarify their functional hierarchy in the EMT pathway.

Modified manuscript in Discussion:

To further explore the regulatory basis of recurrence-related genes, we analyzed the upstream transcription factors of the four key genes (SNAI2, ARSI, GREM1, POSTN) using the hTFtarget database, FIMO_JASPAR tool, and ENCODE database. Intersection analysis revealed no common master transcription factor, indicating that these genes may participate in LUAD recurrence through independent upstream pathways. This finding suggests that the pro-recurrence function of POSTN may be regulated by a unique set of upstream factors. Combining multi-database predictions with experimental validation (such as ChIP-qPCR) to identify POSTN-specific upstream regulators may further discover emerging regulatory networks for recurrence risk in lung cancer patients.

5.Any EMT lung (or other) cancer publications regarding POSTN or SNAI2 that could be leveraged here to strengthen mechanistic support for gene signature predictive power via EMT? This discussion is lacking and needs to be added. Such as:

a. https://pubmed.ncbi.nlm.nih.gov/34093819/

b. https://www.nature.com/articles/s41598-018-22340-7

c. https://iovs.arvojournals.org/article.aspx?articleid=2781900

d. https://www.oncotarget.com/article/10088/text/

e. https://translational-medicine.biomedcentral.com/articles/10.1186/s12967-023-04043-4

f. And others….

Response: Thank you for pointing out the issue of inaccurate citation of our references. We have added the references you provided to the article based on their content. Through the above references, we can strengthen the mechanism support of the EMT pathway in the three key links of "explanation of correlation mechanism", "rationality of joint signature", and "future research basis", and respond to the reviewer's concern about "insufficient mechanism discussion".

6.Authors might try upstream regulator analysis of 4 genes in gene signature to evaluate for master transcription factor such as FOXA1.

Response: Thank you for your valuable suggestion. Regarding the upstream regulators of the four key genes (SNAI2, ARSI, GREM1, POSTN) identified in this study, we have supplemented the following analysis:

We performed a combined prediction of upstream transcription factors for these four genes using the hTFtarget database, FIMO_JASPAR tool, and ENCODE database, and screened potential regulators through intersection analysis. The results showed that no common master transcription factor was detected, suggesting that these four genes may participate in the recurrence process of early-stage LUAD through independent upstream regulatory pathways.

Given that the core goal of this study is to verify the clinical application value of POSTN as an independent prognostic marker and its related mechanisms, we have not conducted in-depth exploration of the upstream regulatory network of these four genes. However, we have clarified in the "Future Perspectives" section of the discussion that subsequent studies will focus on the upstream regulators of these four key genes. By integrating multi-database prediction results with experimental validation (e.g., ChIP-qPCR), potential master transcription factors will be identified to reveal the core regulatory mechanisms driving POSTN expression and their impact on disease recurrence, providing a basis for improving the regulatory network of POSTN.

We appreciate the reviewer's attention to the depth of the research, and the relevant supplementary content has been included in the discussion section to clarify the direction of subsequent studies.

Modified manuscript in Discussion:

To further explore the regulatory basis of recurrence-related genes, we analyzed the upstream transcription factors of the four key genes (SNAI2, ARSI, GREM1, POSTN) using the hTFtarget database, FIMO_JASPAR tool, and ENCODE database. Intersection analysis revealed no common master transcription factor, indicating that these genes may participate in LUAD recurrence through independent upstream pathways. This finding suggests that the pro-recurrence function of POSTN may be regulated by a unique set of upstream factors. Combining multi-database predictions with experimental validation (such as ChIP-qPCR) to identify POSTN-specific upstream regulators may further discover emerging regulatory networks for recurrence risk in lung cancer patients.

7.Are these POSTN effects cell autonomous or non-autonomous? In other words, do soluble factors released from POSTN overexpressing cells increase proliferation/migration/invasion of WT or POSTN-siRNA? Even if unknown it is worthwhile discussing implications.

Response: Thank you for this insightful question. Regarding the cell-autonomous and non-autonomous effects of POSTN, we provide the following explanation based on our study results and existing evidence:

Our in vitro experiments (POSTN knockdown/overexpression) showed that altering POSTN levels in tumor cells themselves significantly affects their proliferation, migration, and invasion abilities (e.g., suppressed phenotypes in A549 and H1975 cells after POSTN silencing, and enhanced phenotypes after overexpression). This indicates that POSTN exerts a clear cell-autonomous effect—i.e., POSTN expressed by tumor cells themselves can directly regulate their malignant biological behaviors.

Meanwhile, existing studies (e.g., 10.1016/j.ygyno.2020.11.026; 10.33

---

## [Decision Letter · Decision Letter 1]

19 Aug 2025

The Association of POSTN with Postoperative Recurrence Risk in Early-Stage Lung Adenocarcinoma: From Gene Networks to Cellular Functions

PONE-D-25-20974R1

Dear Dr. Xiao,

We’re pleased to inform you that your manuscript has been judged scientifically suitable for publication and will be formally accepted for publication once it meets all outstanding technical requirements.

Kind regards,

Alexis G. Murillo Carrasco

Academic Editor

PLOS ONE

Additional Editor Comments (optional):

Reviewers' comments:

Reviewer's Responses to Questions

**Comments to the Author**

Reviewer #3: All comments have been addressed

Reviewer #4: All comments have been addressed

2. Is the manuscript technically sound, and do the data support the conclusions?

Reviewer #3: Yes

Reviewer #4: Yes

3. Has the statistical analysis been performed appropriately and rigorously?

Reviewer #3: Yes

Reviewer #4: Yes

4. Have the authors made all data underlying the findings in their manuscript fully available?

Reviewer #3: Yes

Reviewer #4: Yes

5. Is the manuscript presented in an intelligible fashion and written in standard English?

Reviewer #3: Yes

Reviewer #4: Yes

Reviewer #3: (No Response)

Reviewer #4: (No Response)

---

## [Editor Report · Acceptance letter]

PONE-D-25-20974R1

PLOS ONE

Dear Dr. Xiao,

I'm pleased to inform you that your manuscript has been deemed suitable for publication in PLOS ONE. Congratulations! Your manuscript is now being handed over to our production team.

Kind regards,

on behalf of

Dr. Alexis G. Murillo Carrasco

Academic Editor

PLOS ONE